# Genomic Analysis Revealed the International and Domestic Transmission of Carbapenem-Resistant *Klebsiella pneumoniae* in Chinese Pediatric Patients

Muxiu Jiang,[a] Heng Li,[b,c] Xiao Liu,[b] Nan Shen,[a] Yuanjie Zhou,[a] Wenting Song,[a] Xing Wang,[d] Qing Cao,[a] Zhemin Zhou[b,c]

aDepartment of Infectious Disease, Shanghai Children's Medical Center, School of Medicine, Shanghai Jiao Tong University, Shanghai, China
bPasteurien College, Suzhou Medical College of Soochow University, Suzhou, Jiangsu, China
cSuzhou Key Laboratory of Pathogen Bioscience and Anti-infective Medicine, Suzhou, Jiangsu, China
dDepartment of Laboratory Medicine, Shanghai Children's Medical Center, School of Medicine, Shanghai Jiao Tong University, Shanghai, China

Muxiu Jiang and Heng Li contributed equally to this article. Author order was determined on the basis of type of contribution.

**ABSTRACT** Carbapenem-resistant *Klebsiella pneumoniae* (CRKP) poses a severe threat to public health worldwide. Based on the genomic analysis of 198 CRKP isolates collected at Shanghai Children's Medical Center over the last 8 years (2013 to 2021), we reported the clinical risk, genetic diversity, and prevalence of antimicrobial resistance (AMR) of CRKP in pediatric patients at the genomic level. We found that the $bla_{NDM}$ genes were the predominant carbapenemase genes, followed by $bla_{KPC-2}$ and $bla_{IMP}$. All of the carbapenemases were disseminated mainly by four main types of plasmids, among which one plasmid was associated with a higher risk of bloodstream infections. Notably, we tracked disease outbreaks caused by recent introductions of ST14 CRKP from southeast Asia or western countries, and we reported frequent, repetitive introductions of ST11 from other domestic hospitals that were associated interhospital movement of the patients. The cocirculation of *K. pneumoniae* and AMR plasmids in hospitals highlights the importance of genome sequencing for monitoring and controlling CRKP infections.

**IMPORTANCE** Carbapenem-resistant *Klebsiella pneumoniae* (CRKP) infection in pediatric patients differs from that in adults patients in terms of both genetic and phenotypic features, which remain to be elucidated. We present a summary of prevalent CRKP isolates from Chinese pediatric patients over 8 years, demonstrating the prevalence and clinical importance of New Delhi metallo-$\beta$-lactamase genes in pediatric patients, mainly describing the genomic features of two predominant CRKP clones (ST11 and ST14) in Chinese children, and identifying four carbapenemase-encoding plasmids that contribute to the transmission of most carbapenemase genes in hospitals. Overall, our research provides valuable information about the international and domestic transmission of CRKP isolates that are prevalent in Chinese children and shows the urgent need for genome sequencing-based surveillance systems for monitoring the transmission of CRKP.

**KEYWORDS** carbapenem resistance, *Klebsiella pneumoniae*, whole-genome sequencing, plasmid, child

The emergence and spread of carbapenem-resistant *Klebsiella pneumoniae* (CRKP) in clinics are considered "urgent threats" to public health due to the lack of effective antibiotics (1). CRKP infections frequently occur in patients who stay in intensive care units (ICUs) for long periods and can lead to poor prognosis and high mortality rates of up to 40 to 50% worldwide (2, 3). According to the China Antimicrobial Resistance Surveillance Trial Program, the frequency of CRKP clinical isolation has rapidly increased from 2.9% in 2005 to 24.4% in 2021 in general (https://www.chinets.com) and to 24.7% in 2018 in

Address correspondence to Zhemin Zhou, zmzhou@suda.edu.cn, Qing Cao, caoqing@scmc.com.cn, or Xing Wang, wx_5166@163.com.

The authors declare no conflict of interest.

children (4). The $bla_{KPC}$ and $bla_{OXA}$ genes are the most prevalent carbapenemase genes in CRKP. However, an increasing prevalence of New Delhi metallo-$\beta$-lactamase (NDM) has been reported over the last decade, accounting for 11 and 30% of cases of carbapenem resistance in Europe (5) and China (6), respectively. Furthermore, $bla_{NDM}$ has become the primary carbapenemase in pediatric patients in China and many other countries (6–8).

The genotypes of *K. pneumoniae* strains are currently cataloged by a multilocus sequence typing (MLST) (9). Based on sequence variations in seven housekeeping genes, *K. pneumoniae* have been classified into hundreds of sequence types (STs) that each approximate a group of genetically related strains (9, 10). Mobile genetic elements such as plasmids mediate the transfer of many virulence genes and antimicrobial genes, i.e., carbapenemases, across different STs or from other *Enterobacteriaceae* species (11, 12), which has led to the emergence of a prevalent population worldwide. For example, *K. pneumoniae* ST14 is one of the most prevalent causative agents of nosocomial infections in many countries, and its plasmids encode multiple $\beta$-lactamases, such as those in KPC, CTX-M, SHV, TEM, FOX, CMY, and OXA families (13), as well as NDM genes (14). This ST type, however, has only been described in Chinese individuals in a few case reports (8, 15), whereas ST11 KPC-2 is contemporarily the predominant clone, accounting for >80% of CRKP infections in adults (16).

Comparative genomics of CRKP strains has provided in-depth insights into the evolution of specific CRKP clones in adults (11, 17). Nonetheless, few studies have characterized the transmission dynamics and genomic features of CRKP isolates from pediatric patients. To elucidate the clinical and genome sequence of CRKP infections in Chinese children, we sequenced and analyzed 198 clinical CRKP isolates from patients in Shanghai Children's Medical Center over 8 years and investigated their resistance spectrum, virulence genes, and evolutionary characteristics.

## RESULTS

**Clinical background and genome features of CRKP isolates.** A total of 178 patients were enrolled between January 2013 and August 2021 at Shanghai Children's Medical Center (SCMC), which is a national medical center for treating pediatric patients from all over China. Most of the enrolled patients were infants (aged ≤1 year) (79.8%, 142/178), admitted to the intensive care unit (ICU) (75.8%), had underlying congenital heart disease (69, 38.8%), were born prematurely (30, 16.9%), or received liver transplantation or prior Kasai's surgery (28, 15.7%) (Table 1). Over one-fifth (36, 20.2%) of the patients died or left the hospital before recovery, among whom 50% of patients (18/36) had confirmed CRKP bloodstream infections.

A total of 198 isolates were collected, mainly from sputum (96, 48.4%) and blood (73, 36.8%) samples. Based on the sequences of housekeeping genes, we assigned each CRKP isolate to one of 58 STs in four subspecies: *K. pneumoniae* (179/198, 90.4%), *K. quasipneumoniae* subsp. *similipneumoniae* (14/198, 7.1%), *K. quasipneumoniae* subsp. *quasipneumoniae* (3/198, 1.5%), and *K. variicola* subsp. *variicola* (2/198, 1.0%) (Fig. 1A). ST11 and ST14 were the predominant sequence types, accounting for 26.3% (52/198) and 16.7% (33/198) of the isolates, respectively. NDM variants (96, 48.5%) were the primary carbapenemases carried by 96 isolates from 37 STs, followed by KPC-2 (53, 26.7%), which was mostly from ST11, and IMP variants (33, 16.7%) from 19 STs (Fig. 1B and Table 1). We also found that eight isolates did not encode any known carbapenemase, four isolates encoded both KPC-2 and NDM-1, and four isolates encoded both NDM-1 and IMP-4. Notably, we obtained 19 sets of two to three CRKP isolates each from the same patient. Some of these isolates were isolated from different parts of the body, or from samples that were collected 1 to 13 months apart. Except for two sets, isolates from the same patient were always assigned the same ST and exhibited no or very few variations in their core genomes (see Data Set S2 in the supplemental material).

**Evaluation of AMR genotypes and phenotypes of CRKP isolates.** To evaluate the association between antimicrobial resistance (AMR) genotypes and phenotypes, antimicrobial susceptibility testing was carried out on 73 isolates from the bloodstream samples (Fig. 1D; see also Data Set S1). The antimicrobial assays showed that all isolates

**TABLE 1** Clinical and molecular characteristics of carbapenem-resistant *K. pneumoniae* infections in Chinese children[a]

| Characteristics | Clinical and microbial data |
|---|---|
| Clinical characteristics | |
| Patients (*n*) | 178 |
| Median age (IQR) in mos. | 3.5 (0.6–8.5) |
| No. (%) of subjects by age | |
| <1 mo | 58 (32.6) |
| 1–6 mo | 57 (32.0) |
| 6–12 mo | 27 (15.2) |
| >12 mo | 36 (20.2) |
| No. (%) of male patients | 110 (61.8) |
| No. (%) of ward patients | |
| ICU | 135 (75.8) |
| PICU | 52 (29.2) |
| NICU | 31 (17.4) |
| CICU | 52 (29.2) |
| Hematology | 7 (3.9) |
| Neonatology | 14 (7.8) |
| Pneumology | 13 (7.3) |
| Others | 9 (5.1) |
| No. (%) of patients with underlying disease | |
| Premature | 30 (16.9) |
| Congenital heart disease | 69 (38.8) |
| Liver transplantation or Kasai's surgery | 28 (15.7) |
| Hematologic disease | 13 (7.3) |
| Pulmonary disease | 16 (8.9) |
| Nervous system disease | 8 (4.4) |
| Others | 14 (7.9) |
| No. (%) of patients | |
| Previous hospitalization within 3 mo | 95 (53.4) |
| Prior surgery within 3 mo | 92 (51.6) |
| Clinical outcomes, no. (%) | |
| Cure | 141 (79.2) |
| Given up | 20 (11.2) |
| Death | 16 (9.0) |
| 14-day crude mortality | 11 (6.2) |
| 90-day crude mortality | 16 (9.0) |
| | |
| Microbial characteristics | |
| No. of isolates | 198 |
| Resources, no. (%) of isolates | |
| Blood | 73 (36.8) |
| Sputum | 96 (48.4) |
| Others | 29 (14.6) |
| Carbapenemase, no. (%) | |
| NDM[b] | 96 (48.5) |
| KPC-2 | 53 (26.7) |
| IMP | 33 (16.7) |
| IMP-4+NDM-1 | 4 (2.0) |
| KPC-2+NDM-5 | 4 (2.0) |
| None | 8 (4.0) |
| Sequence type, no. (%) | |
| ST11 | 52 (26.3) |
| ST14 | 33 (16.7) |
| ST433 | 7 (3.5) |
| ST17 | 5 (2.5) |
| ST20 | 6 (3.0) |
| ST1308 | 5 (2.5) |
| ST15 | 5 (2.5) |
| ST76 | 5 (2.5) |
| Other STs | 80 (40.4) |

[a]IQR, interquartile range; ICU, intensive care unit; PICU, pediatric ICU; NICU, neonatal ICU; CICU, cardiac ICU; NDM, New Delhi metallo-$\beta$-lactamase; KPC, *K. pneumoniae* carbapenemase; IMP, imipenemase; ST, sequence type.
[b]Including NDM-1 (*n* = 72, 36.3%), NDM-5 (*n* = 22, 11.1%), NDM-7 (*n* = 1, 0.5%), and NDM-9 (*n* = 1, 0.5%).

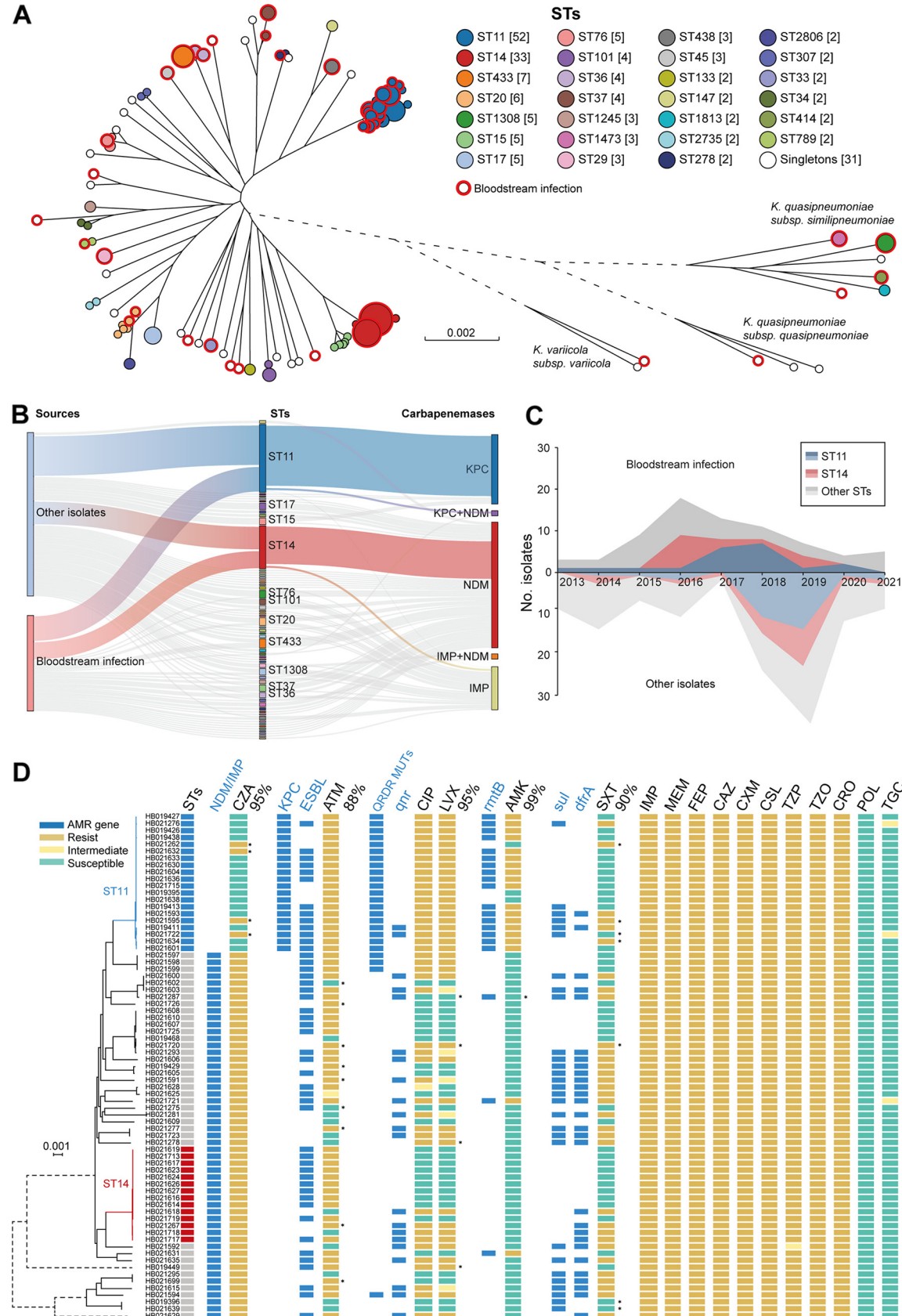

**FIG 1** Characteristics of CRKP from pediatric patients in the SCMC during 2013 to 2021. (A) A maximum-likelihood phylogeny of all 198 SCMC isolates. Nodes were color-coded by their diverse STs (as indicated in the key). Branches connecting different subspecies

**TABLE 2** Antimicrobial susceptibility testing results for 73 carbapenem-resistant *K. pneumoniae* isolates[a]

| Agent | All CRKP (*n* = 73) | | | | KPC-2 producers (*n* = 20) | | | | MBL producers (*n* = 53) | | | | *P*[b] |
|---|---|---|---|---|---|---|---|---|---|---|---|---|---|
| | $MIC_{50}$ | $MIC_{90}$ | R% | S% | $MIC_{50}$ | $MIC_{90}$ | R% | S% | $MIC_{50}$ | $MIC_{90}$ | R% | S% | |
| IMP | 64 | 128 | 100 | 0 | 64 | 128 | 100 | 0 | 32 | 128 | 100 | 0 | |
| MEM | >64 | >64 | 100 | 0 | >64 | >64 | 100 | 0 | 64 | >64 | 100 | 0 | |
| FEP | >128 | >128 | 100 | 0 | >128 | >128 | 100 | 0 | >128 | >128 | 100 | 0 | |
| CAZ | >32 | >32 | 100 | 0 | >32 | >32 | 100 | 0 | >32 | >32 | 100 | 0 | |
| CXM | >32 | >32 | 100 | 0 | >32 | >32 | 100 | 0 | >32 | >32 | 100 | 0 | |
| CSL | >128 | >128 | 100 | 0 | >128 | >128 | 100 | 0 | >128 | >128 | 100 | 0 | |
| CZO | >32 | >32 | 100 | 0 | >32 | >32 | 100 | 0 | >32 | >32 | 100 | 0 | |
| CRO | >32 | >32 | 100 | 0 | >32 | >32 | 100 | 0 | >32 | >32 | 100 | 0 | |
| TZP | >256 | >256 | 99 | 0 | >256 | >256 | 100 | 0 | >256 | >256 | 98 | 0 | |
| CZA | >64 | >64 | 78 | 22 | 2 | >64 | 20 | 80 | >64 | >64 | 100 | 0 | <0.001 |
| ATM | >128 | >128 | 81 | 18 | >128 | >128 | 100 | 0 | >128 | >128 | 74 | 25 | 0.033 |
| CIP | 2 | >8 | 60 | 38 | >8 | >8 | 100 | 0 | 0.25 | >8 | 45 | 53 | <0.001 |
| LEV | 2 | >16 | 53 | 40 | >16 | >16 | 100 | 0 | 0.25 | >8 | 36 | 55 | <0.001 |
| POL | 0.5 | 1 | 0 | 100 | 0.5 | 0.5 | 0 | 100 | 0.5 | 1 | 0 | 100 | |
| TGC | 0.5 | 2 | 0 | 96 | 1 | 4 | 0 | 90 | 0.5 | 2 | 0 | 98 | |
| SXT | 0.5 | >32 | 45 | 55 | ≤0.25 | >32 | 30 | 70 | 2 | >32 | 51 | 49 | 0.109 |
| AMK | ≤1 | >128 | 26 | 74 | >128 | >128 | 85 | 15 | ≤1 | 2 | 4 | 96 | <0.001 |

[a]MICs are expressed as mg/L. CRKP, carbapenem-resistant *K. pneumoniae*; KPC, *K. pneumoniae* carbapenemase; MBL, metallo-$\beta$-lactamase; $MIC_{50}$/$MIC_{90}$, 50%/90% MIC; R%, percentage of resistant isolates; S%, percentage of susceptible isolates; IMP, imipenem; MEM, meropenem; FEP, cefepime; CAZ, ceftazidime; CXM, cefuroxime; CSL, cefoperazone-sulbactam; CZO, cefazolin; CRO, ceftriaxone; TZP, piperacillin-tazobactam; CZA, ceftazidime-avibactam; ATM, aztreonam; CIP, ciprofloxacin; LEV, levofloxacin; POL, polymyxin B; TGC, tigecycline; SXT, trimethoprim-sulfamethoxazole; AMK, amikacin; R% are indicated in boldface.
[b]*P* values were calculated as R%$_{KPC-2}$/R%$_{MBL}$.

were resistant to $\beta$-lactams and their corresponding inhibitors, including imipenem, meropenem, cefepime, ceftazidime, cefuroxime, cefoperazone-sulbactam, cefazolin, and ceftriaxone (Table 2). Various levels of susceptibilities to aztreonam (ATM) (18%), ceftazidime-avibactam (CZA) (22%), ciprofloxacin (38%), levofloxacin (40%), trimethoprim-sulfamethoxazole (55%), amikacin (74%), tigecycline (96%), and polymyxin B (100%) were observed (Table 2). We managed to predict the results of antimicrobial assays based on AMR genotypes with 88 to 99% consistency (Fig. 1D). Ceftazidime-avibactam and amikacin resistances were associated with the presence of metallo-$\beta$-lactamase (MBL) carbapenemases (95% consistency) and *rmtB* (99%), respectively. Aztreonam and trimethoprim-sulfamethoxazole resistance were each associated with two *bla*$_{KPC-2}$ and *bla*$_{CTX-M}$ genes (88%) and *sul* and *dfrA* genes (90%). Finally, quinolone resistance was associated with *qnr* genes, as well as quinolone resistance-determining region mutations in the *gyrA* and/or *parC* genes (95%).

**Prediction of dominant plasmids in CRKP isolates associated with horizontal gene transfer.** To investigate the plasmid vectors that were responsible for the transmission of carbapenemases, we aligned all the assembled genomes with 6,779 *Klebsiella* plasmids deposited in RefSeq (Fig. 2A) using BLASTn and identified four primary carbapenemase-encoding plasmids in the SCMC isolates. These plasmids were named as pSCMC1 (similar to MZ156801; IncX3), pSCMC2 (CP082020; IncFII), pSCMC3 (CP039821; IncN), and pSCMC4 (CP003684; IncU) for convenience of description. At least one of the four plasmids was detected in 92% (67/73) of bloodstream isolates and 82% (102/125) of other sources of isolates. We reconstructed a maximum-likelihood

**FIG 1 Legend (Continued)**
were shortened and highlighted in dashed lines. The CRKPs from bloodstream (red circle) were found in diverse STs and subspecies. (B) Sankey plot showing the association between carbapenemases and STs from bloodstream and other sources. (C) The frequencies (*y*-axis) of finding ST11, ST14, and other STs in bloodstream (top) and other samples (bottom) for each year (*x*-axis) during 2013 to 2021. (D) Heat plot (right) that shown phenotypic (yellow to green boxes) and genotypic (blue boxes) antimicrobial resistance identified among 73 bloodstream isolates, which are ordered along the phylogeny (left) as extracted from panel A. ST, sequence type; KPC-2, *K. pneumoniae* carbapenemase; NDM, New Delhi metallo-$\beta$-lactamase; IMP, imipenemase; AMR, antimicrobial resistance; CZA, ceftazidime-avibactam; ESBL, extended-spectrum $\beta$-lactamase; ATM, aztreonam; QRDR, quinolone resistance-determining regions; CIP, ciprofloxacin; LVX, levofloxacin; AMK, amikacin; SXT, trimethoprim-sulfamethoxazole; IPM, imipenem; MEM, meropenem; FEP, cefepime; CAZ, ceftazidime; CXM, cefuroxime; CSL, cefoperazone-sulbactam; TZP, piperacillin-tazobactam; CZO, cefazolin; CRO, ceftriaxone; POL, polymyxin B; TGC, tigecycline.

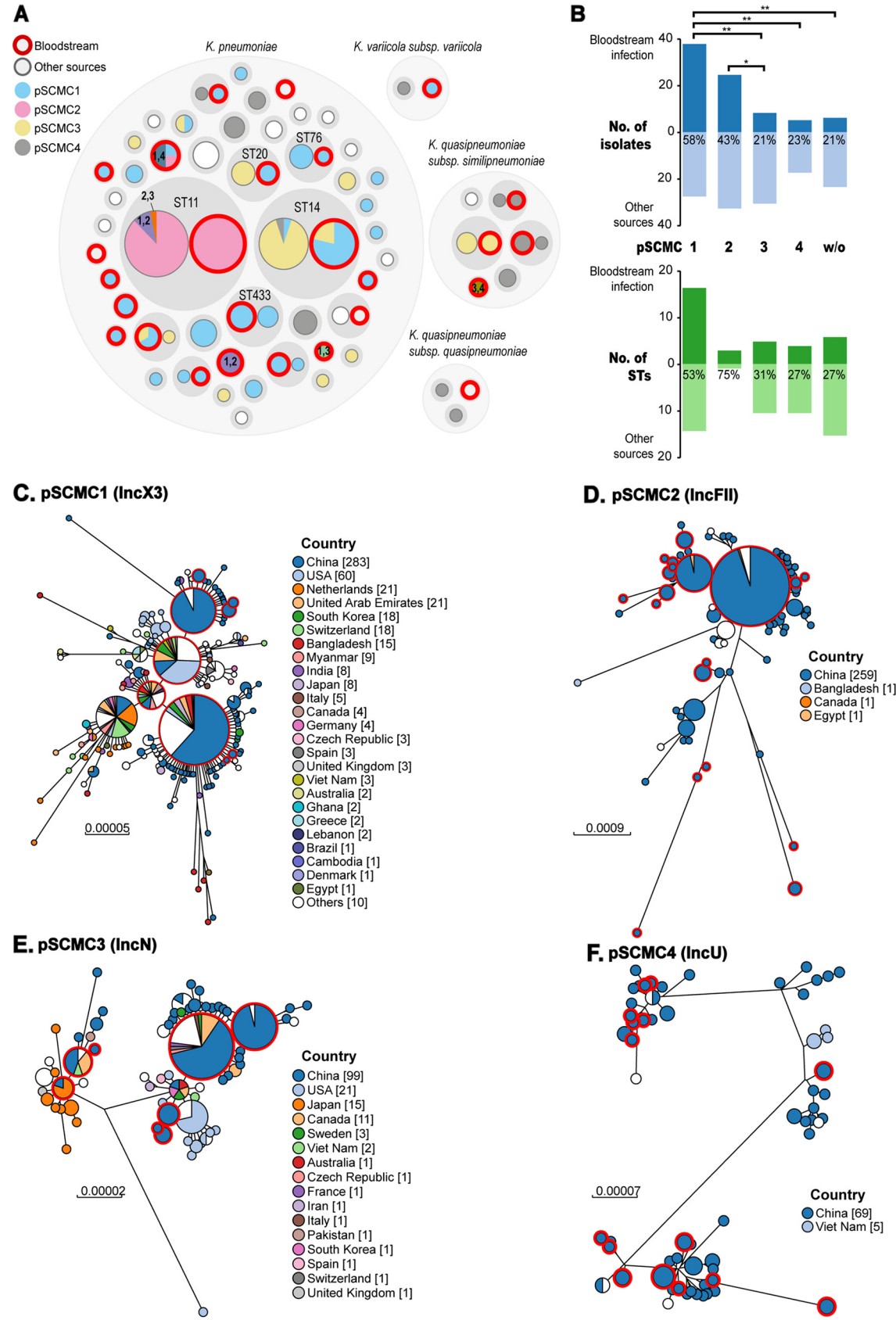

**FIG 2** Four predominant pSCMC plasmids that were responsible for most of carbapenemases in the SCMC. (A) A hierarchical bubble plot shows the distribution of pSCMC plasmids in varied STs. The CRKP from bloodstream are highlighted with red circles.

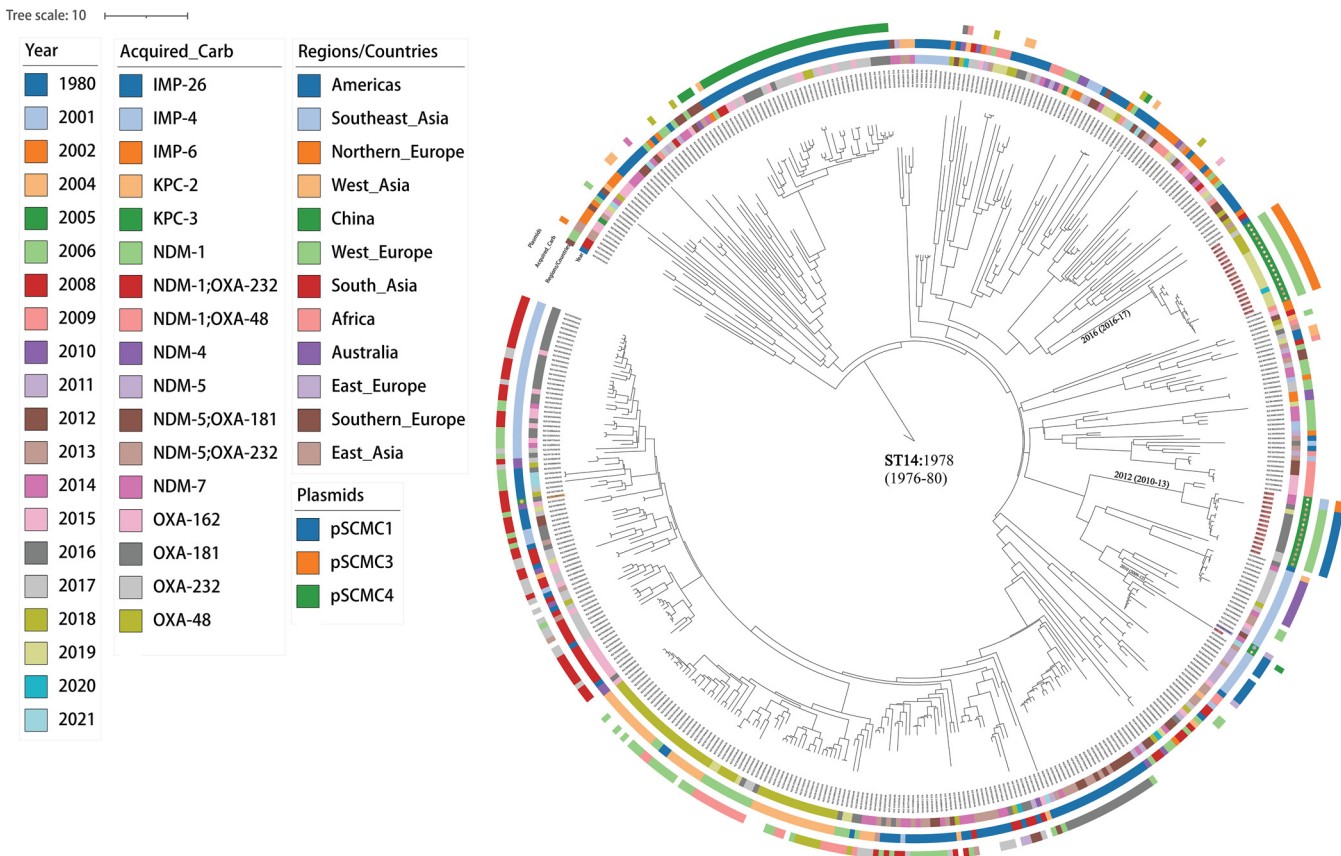

**FIG 3** The maximum clade credibility (MCC) trees for genomes of ST14 with 33 Shanghai variants and 444 global isolates from EnteroBase. The ST14 isolates were color-coded according to the isolation year of strains, the geographical origin of the genomes, and their carried carbapenem resistance genes. Isolates from present study were marked with solid (bloodstream) or hollow (other sources) stars in the middle circle of the tree. The other two ST14 isolates from Jiangxi (2014) and Hong Kong (2016), China, are marked with purple and orange solid circles, respectively. Plasmids associated with SCMC isolates also shown. ST, sequence type; SCMC, Shanghai Children's Medical Center; ST, sequence type; IMP, imipenemase; KPC, *K. pneumoniae* carbapenemase; NDM, New Delhi metallo-β-lactamase; OXA, oxacillinase.

tree for each predominant plasmid using both SCMC plasmids and similar sequences from RefSeq. The results showed that pSCMC1 and pSCMC3 were found internationally, whereas pSCMC2 and pSCMC4 were largely restricted to China (Fig. 2C to F). In addition, most plasmids carried diverse carbapenemases and were found in diverse STs or species, except that pSCMC2 was mostly associated with KPC-2 genes in ST11 (Fig. 2A; see also Fig. S1). Notably, we found that 58% (37/64) of pSCMC1 plasmids were from bloodstream samples, which was significantly higher than the frequencies of pSCMC3 and pSCMC4 being present in bloodstream samples ($P < 0.01$) (Fig. 2B, top). The association between the pSCMC1 plasmids and bloodstream infections was independent of its bacterial host, since the bloodstream isolates that carried pSCMC1 plasmids were from at least 17 different STs (Fig. 2B, bottom).

**International introduction of ST14-producing *K. pneumoniae* isolates.** We compared the genomes of 33 ST14-producing CRKP isolates with a global collection of 444 ST14 *K. pneumoniae* genomes deposited in EnteroBase (see Fig. S2A). A maximum clade credibility (MCC) tree (Fig. 3) was generated using BEAST based on 20,672 single nucleotide polymorphisms in the nonrepetitive, nonrecombinant core genome. The

**FIG 2** Legend (Continued)

(B) Percentages refer to the number of isolates (top) and STs (bottom) from bloodstream to all sources isolates for each pSCMC plasmid. *, $P < 0.05$; **, $P < 0.01$. (C to F) Maximum-likelihood phylogeny tree of the four predominant plasmids from SCMC isolates (in circles with red halos) or from RefSeq. The circles are sized relative to the numbers of associated plasmids and color-coded by the geographic sources of the plasmids. CRKP, carbapenem-resistant *K. pneumoniae*; SCMC, Shanghai Children's Medical Center; ST, sequence type.

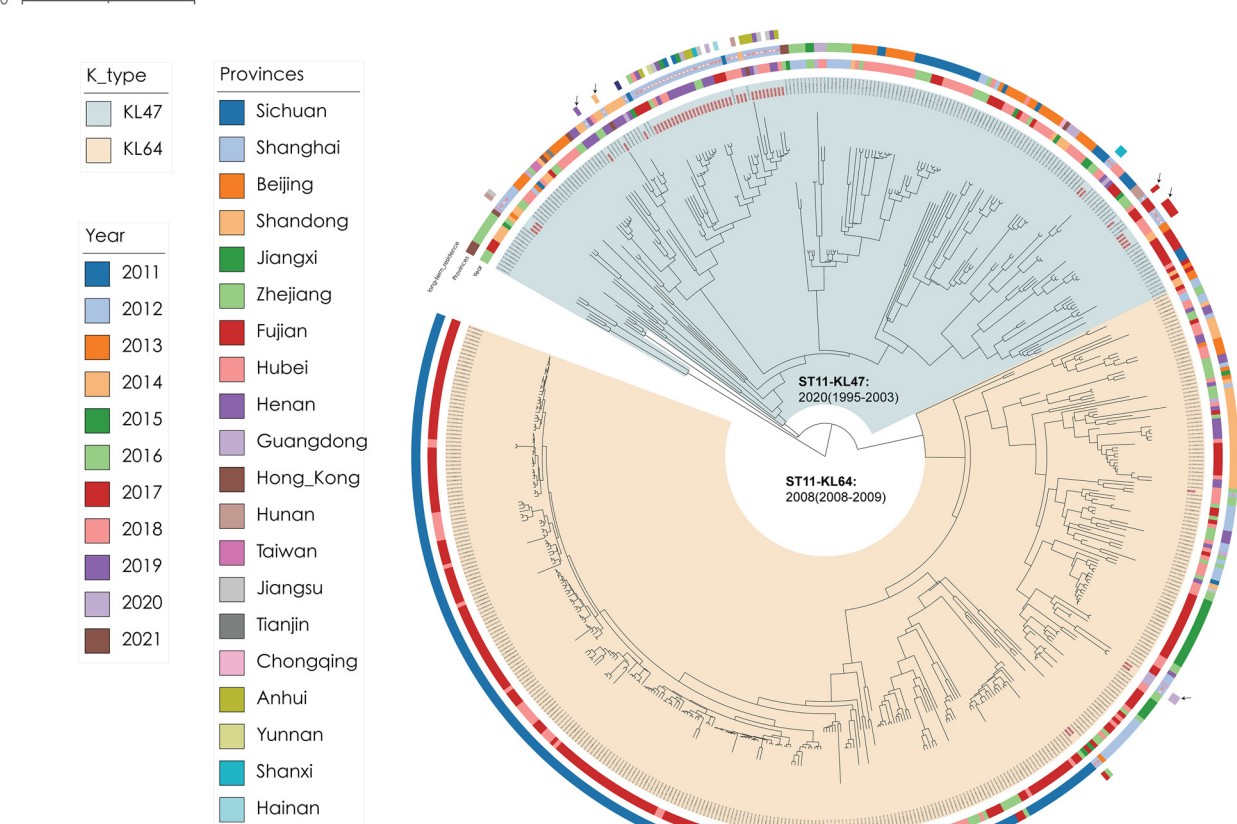

**FIG 4** Maximum clade credibility trees for genomes of ST11 with 52 isolates from SCMC plus 577 publicly available Chinese genomes from EnteroBase. The ST11 isolates are color-coded according to K antigen, the isolation year of strains, and the geographical origin of the genomes. Isolates from the present study are marked with solid (bloodstream) or hollow (other sources) stars in the middle circle of the tree. The patient's long-term residences are also shown. ST, sequence type; SCMC, Shanghai Children's Medical Center.

results showed that the most recent common ancestor (MRCA) of all ST14 isolates originated in Northern Europe in 1978 (95% confidence interval [CI] = 1976 to 1980) and was globally transmitted soon after. However, ST14 was not introduced into China until 2010, when it was transmitted to Jiangxi and later to Shanghai from Southeast Asia. A second introduction of ST14 from Southeast Asia into Shanghai occurred in 2012. The introduced bacteria carried a pSCMC3-IMP-4 plasmid when it was first isolated in 2014, and later, it acquired a pSCMC1-NDM-1 plasmid that led to multiple bloodstream infections after 2016. The last introduction of ST14 from North America occurred in 2016, and resulted to a bunch of isolates between 2018 and 2019 (Fig. 3). Although they cause multiple bloodstream infections and deaths, as shown here, none of the ST14 isolates in the present study carried *iro*, *rmpA*, *rmpA2*, or *iuc*, and were therefore not considered hypervirulent clones (18) (see Data Set S1).

**Domestic transmission of ST11 KPC-2-producing *K. pneumoniae* in China.** We compared all 52 ST11 KPC-2-producing CRKP isolates from SCMC with 577 publicly available Chinese ST11 *K. pneumoniae* isolate genomes in EnteroBase, and we found that ≥92% of Chinese isolates fell into one of two closely related clades in the tree (see Fig. S2B). We denoted these two clades ST11-KL47 and ST11-KL64 according to the *in silico* predictions of their somatic antigens. BEAST analysis showed that MRCA of ST11-KL47 originated in 2000 (95% CI = 1995 to 2003) and that its K antigen changed to KL64 in 2008 (95% CI = 2008 to 2009). All 52 ST11 isolates from SCMC formed 10 clusters in the MCC tree. Most SCMC clusters were very small, consisting of no more than 4 isolates, except that there was a large cluster of 35 isolates that were continuously identified from 2015 to 2020 (Fig. 4). According to the tree, isolates have been

frequently transferred between provinces, and many SCMC isolates were recently transferred from Fujian, Guangdong, Shandong, Henan, and other areas of China. Such frequent transmissions might be partially associated with interhospital admissions of patients, as 40% (21/52) of the ST11 isolates from SCMC were collected from patients who experienced surgeries in other hospitals, compared to only 11% of non-ST11 isolates that were collected from patients with a history surgery in other hospitals ($P < 0.001$). Nine of the surgery-related ST11-KPC-2 isolates were collected from patients with bloodstream infections between 2017 and 2018, which may explain the excessive isolation of ST11 at SCMC during the time period (Fig. 1C; see also Data Set S1).

In addition, virulence gene analysis showed that all 52 ST11 isolates harbored *ybt* on an ICEKp3 element. Three ST11-KL64 isolates carried a KpVP-1 plasmid harboring the *rmp1*, *rmpA2* and *iuc1* genes, and two ST11-KL47 isolates, namely, HB019466 and HB021276, carried the *iuc1* and *iuc5* genes, respectively (see Data Set S1). In addition, four ST11-KPC-2 CRKP isolates were resistant to ceftazidime-avibactam with high MIC values (see Data Set S1).

## DISCUSSION

All the CRKP strains in the present study were phenotypically or genotypically multi-drug-resistant strains and were broadly resistant to commonly used antibiotics, including third-generation cephalosporins, aztreonam, and several cases of fourth-generation cephalosporins. We showed that genomic sequencing could aid the treatment of such isolates because it allowed a rapid prediction of AMR profiles with high accuracy, which led to correct use of antimicrobials or antimicrobial combinations for treatment and contribute to a better outcome for patients (19). There were a few cases for each drug in which genomic-based predictions were inconsistent with antimicrobial assays. These might indicate new AMR mechanisms that need to be investigated. In particular, 4 KPC-2 isolates exhibited ceftazidime-avibactam resistance, even though none of the patients had received any ceftazidime-avibactam treatment. Previous studies revealed that ceftazidime-avibactam resistance in KPC-producing isolates was associated with point mutations, increased gene copy number, and/or overexpression of the $bla_{KPC-2}$ gene (20–22). However, we found that there was no additional point mutation or higher copy numbers of KPC-2 in the ceftazidime-avibactam-resistant isolates compared to their genetically related, susceptible neighbors. Therefore, the mechanism underlying the development of ceftazidime-avibactam resistance in pediatric patients needs to be elucidated.

Thousands of AMR plasmids have been identified in *Klebsiella pneumoniae*, especially those of the IncX3, IncN, and IncFII types that are often associated with carbapenemases, extended-spectrum $\beta$-lactamases, and AmpC $\beta$-lactamases and are of high clinical importance (23, 24). We found that similar to the bacterial host, the AMR plasmids identified in pediatric hospitals also followed the Zipf distribution with four primary plasmids accounting for 80 to 90% of infections. We reconstructed the sequences of four dominant plasmids in this study and compared each plasmid to genetically similar plasmids from the NCBI RefSeq database to estimate a maximum-likelihood phylogeny. The results showed that the plasmids fell into two categories of "dispersed" and "restricted" in terms of both host and geographic distributions. pSCMC1 (IncX3) and pSCMC3 (IncN) both carried mostly $bla_{NDM}$ genes and had "dispersed" host and country sources (Fig. 2C to F; see also Fig. S3). pSCMC1 was carried by bacteria from 25 *Enterobacteriaceae* species in 35 countries worldwide (Fig. 2C), and pSCMC3 (IncN) plasmids were found in 23 *Enterobacteriaceae* species from 16 countries (Fig. 2E; see also Fig. S3). Both plasmids are very frequently transmitted between different hosts and different countries, leading to reports of almost identical genetic plasmids from multiple species in many countries (Fig. 2C and E; see also Fig. S3). As a result, it is impractical to estimate the origin of a plasmid using the sequences alone. In contrast, pSCMC2 (IncFII) and pSCMC4 (IncU) were "restricted" and mostly only found in *Klebsiella* isolates from China (Fig. 2D and F; see also Fig. S3). Most pSCMC4 plasmids

carried $bla_{NDM}$, whereas pSCMC2 plasmids carried mostly $bla_{KPC-2}$ genes and were associated with the national dissemination of ST11-KPC-2 CRKPs (25).

It is very concerning that almost identical plasmids could be isolated from epidemiologically unrelated sources. Previous studies also reported the identification of genetically closely related plasmids between those from humans and those from environments, such as hospitals, farm yards, and aquatic environments (14, 26–28). Therefore, the use of measures other than antimicrobials, e.g., hand hygiene, patient isolation, minimization of physical contact with animals, and disinfection of the environment, may decrease the spread of $bla_{NDM}$ and $bla_{IMP}$ genes among children.

Importantly, we found that some of these plasmids have been transmitted internationally, sometimes even too quickly to result in variation in their conserved regions. Some CRKP lineages, i.e., ST11, carried IncFII-KPC-2 for decades (Fig. 2A; see also Fig. S1B), whereas most lineages, such as ST14, changed the AMR plasmids frequently and could lead to severe bloodstream infections and complications in the clinical choice of antimicrobials (Fig. 2A and B). In the present study, pSCMC1 plasmids were found to be associated with bloodstream infections regardless of their bacterial hosts (Fig. 2B). However, we did not find any known virulence genes or obvious AMR genes, except for carbapenemases in the sequences. Further investigation is therefore needed for this plasmid.

In addition, CRKP isolates that carry multiple carbapenemases in one or multiple plasmids have been widely reported and studied (7, 29, 30). We also found eight isolates that encoded two carbapenemases, either IMP-4 and NDM-1 or KPC-2 and NDM-5. Several studies have revealed that the convergence of $bla_{KPC-2}$ and $bla_{NDM-5}$ or the coexistence of $bla_{IMP-4}$ and $bla_{NDM-1}$ in a single strain has the ability to extend the resistance profile of the bacteria, allowing the bacteria to potentially be transmitted between patients (31–33), posing a significant threat to public health. Therefore, it is necessary to strengthen the surveillance of these isolates among children to improve clinical treatment and management.

ST14 has been one of the predominant CRKP clones in many countries (13, 34). Over the last decade, we identified at least two introductions of ST14 strains into Shanghai from different geographic sources and a third introduction into Jiangxi and Shanghai from Southeast Asia (Fig. 3). The imported bacteria caused an increased number of infections in pediatric patients and survived in China for at least 5 to 10 years until the end of our enrollment. Several other studies also reported nosocomial infections due to ST14-NDM-1 in China after 2016 (8, 15, 35). This highlighted ST14 as a potential emerging pathogen that could lead to more disease outbreaks in pediatric and general hospitals. Notably, bloodstream infections caused by ST14 were associated with the presence of the pSCMC1 plasmid and NDM genes. All three introductions were from clades of ST14 that lacked NDM genes when they were first transmitted into China. However, these strains acquired NDM-1 (Shanghai, twice) or NDM-5 (Jiangxi) genes via the introduction of pSCMC1 or pSCMC3 plasmids shortly after their arrival (Fig. 3). Such functional convergences of *K. pneumoniae* suggested potential selective stress in the acquisition and maintenance of NDM genes in pediatric patients in China. Our study, however, included isolates from only one hospital in East China and was not able to provide an epidemic overview for this potential pathogen and its association with NDM genes.

ST11 is a prevalent clone in Asia, and is associated with multiple carbapenemase genes, including $bla_{KPC-2}$, $bla_{NDM}$, and $bla_{VIM}$ (36). Most ST11 isolates that have been circulating among adults in China carry only the KPC-2 gene and belong to a clonal and genetically distinguished lineage (37). Here, we reconstructed the evolutionary histories of the two most concerning ST11 clades, namely, ST11-KL47 and ST11-KL64, based on >500 Chinese isolates. We identified frequent domestic transmissions of ST11-KPC-2 isolates in China, including at least 10 independent introductions into SCMC since their emergence 20 years ago (Fig. 4). The dissemination of ST11 may have been facilitated by interhospital movements of patients. An ST11 isolate could infect a patient

during his or her stay in one hospital and be transmitted elsewhere when the patient visited another hospital within a short time. To support this assumption, we showed that ~40% of ST11-KPC-2 isolates from SCMC were associated with recent surgeries at other hospitals. Second, nine ST11 isolates in SCMC were clustered with isolates from the same provinces as the patient (Fig. 4). Finally, comparative genomics of multiple isolates from the same patient also showed that CRKP could stay within patients for several months, leaving enough time for interhospital transmission. A nationwide, genome-based surveillance system for *K. pneumoniae* would enable the tracking of patients and allow further evaluation of the assumption.

Furthermore, genetically different strains from the same species could be isolated from the same patient. Such microdiversity is a result of either repetitive infections or *in vivo* diversifying evolution of a single clone during prolonged infections, and it could be clinically important when different strains have different AMR susceptibilities. For example, ST11-KPC-2 isolates have been reported to gain additional resistance to ceftazidime-avibactam and tigecycline by mutations in D179Y and the *ramR* gene during long-term treatments (20, 38). Here, we compared genomic sequences of 19 sets of CRKPs that were each isolated from the same patients. We found repetitive infections of different strains that differed form each other by >3,400 core gene sequence variations and >1,500 gain/loss of accessory genes for two patients (see Data Set S2). Detailed investigation showed that one patient carried different CRKPs in different hospital admissions, and the other gained a second infection during her >1-year stay in SCMC. In contrast, we found only 1.3 (0 to 4) mutations in the core genes in strains isolated from the remaining 17 patients (see Data Set S2), indicating *in vivo* evolution of single infections. Interestingly, much more frequent gene content variations (~31 genes; 0 to 283) were found in their accessory genomes, which were associated with gain/loss of prophages and plasmids. A similar phenomenon has also been reported in *Salmonella*, where a laboratory-leaking strain gained multiple prophages and plasmids without any trackable change in the core genome (39). This also highlighted the roles of AMRs and virulence factors carried by plasmids, prophages and other mobile elements as the main driving forces for *in vivo* adaptation of CRKP.

Furthermore, the overall mortality rate of 178 pediatric patients with CRKP in the present study was 9.0%, while 15.2% of patients with bloodstream infections progressed to death. Approximately only 3% of CRKP isolates in the present study were attributed to so-called "hypervirulence" clones, which were believed to be associated with invasive infections (40–42) due to the presence of the siderophores aerobactin (*iuc*), and salmochelin (*iro*) or hypermucoid-encoding genes (*rmpA* and *rmpA2*). We, however, found no difference between hypervirulent clones and others in terms of bloodstream infection or the mortality rates. Similarly, ST11-KL64 was recently acclaimed to be more virulent than ST11-KL47 and to cause more infections (16, 43). Nonetheless, we found no greater mortality ($P = 0.51$) and 5 times fewer infection rates due to ST11-KL64. Finally, we attributed the high mortality rate to severe underlying diseases of the patients. Many of these patients received liver transplantation or allogeneic hematopoietic stem cell transplantation (allo-HSCT), which may lead to immunocompromised status and a worse outcome during CRKP infections (44).

In conclusion, we present a summary of CRKP in Chinese pediatric patients over 8 years and first describe the genetic background of four carbapenemase-encoding plasmids in pediatric patients. We demonstrated the prevalence and clinical importance of NDM genes, which may be associated with environmental selective stress, in pediatric patients (Fig. 5). pSCMC1-NDM plasmids were carried by genetically diverse populations of *K. pneumoniae*. New pathogens may emerge when they are picked up by *K. pneumoniae* STs that are found worldwide, such as ST14. In contrast, the ST11-KPC-2 isolates, which were prevalent in Chinese adults, mainly led to CRKP infections in pediatric patients with a surgery history in general and external hospitals. Further surveillance is required to monitor and control the transmission of *K. pneumonia* through plasmid-mediated international and domestic introduction.

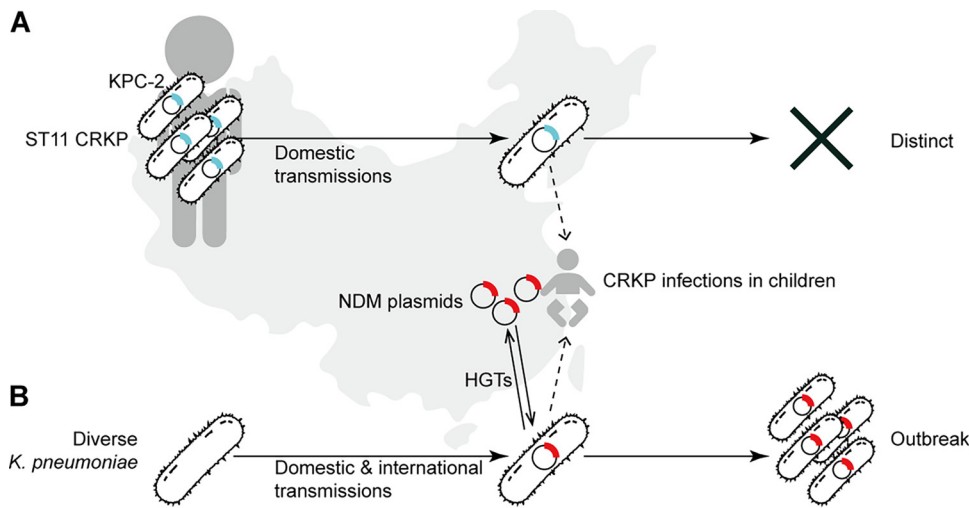

**FIG 5** Cartoon of the dynamics of CRKP in pediatric patients in China. (A) ST11-KPC-2 CRKP was the dominant clone in adults and infected pediatric patients repetitively during surgeries or inpatient stays in general hospitals. These infectious strains however, did not survive in children and died out after a short time period. (B) *K. pneumoniae* with varied genetic backgrounds could also infect pediatric patients after domestic or international transmissions. These isolates might cause disease outbreaks after gaining NDM plasmids that had been circulating among pediatric patients. CRKP, carbapenem-resistant *K. pneumoniae*; ST, sequence type; KPC-2, *Klebsiella pneumoniae* carbapenemase; NDM, New Delhi metallo-$\beta$-lactamase.

## MATERIALS AND METHODS

**Study design and population.** A total of 213 CRKP isolates, *K. pneumoniae* isolates that are resistant to one or more carbapenems, were collected from 178 inpatients in the SCMC from 2013 to 2021. For each patient, we collected clinical metadata related to age, sex, resources, wards, underlying diseases, ICU admission, previous hospitalization, and surgery within 3 months and clinical outcomes of patients and clinical sources of strains. We recorded 14- and 90-day overall mortality rates for the patients who were included in the present study. A total of 198 CRKP isolates from 178 pediatric patients were included in the present study after excluding patients with polymicrobial infections or incomplete medical records (a flow chart diagram is shown in Fig. S4). The Ethics Committee of Shanghai Children's Medical Center exempted this study from review since it focused on bacteria and had no impact on patients.

**Antimicrobial susceptibility testing.** Initially, the disk diffusion assay of meropenem and imipenem (Oxoid) was used for the phenotypic antimicrobial investigations of all the *K. pneumoniae* strains according to Clinical and Laboratory Standards Institute (CLSI) guidelines. The minimum inhibitory concentrations (MIC) values for 17 antimicrobial agents (imipenem, meropenem, ceftazidime-avibactam, cefepime, tigecycline, amikacin, polymyxin B, ceftazidime, cefuroxime, aztreonam, ciprofloxacin, levofloxacin, cefoperazone-sulbactam, piperacillin-tazobactam, trimethoprim-sulfamethoxazole, cefazolin, and ceftriaxone) were determined using the broth microdilution method (Sensitire CHN1GNHH panel; Thermo Fisher Scientific) for 73 CRKP isolates from bloodstream. All the results were interpreted according to Clinical and Laboratory Standards Institute (CLSI; 2022) criteria, except for the results of tigecycline, which were interpreted based on U.S. Food and Drug Administration criteria since there are no CLSI breakpoints for this agent. *Escherichia coli* ATCC 25922 was used as a quality control.

**Whole-genome sequencing and bioinformatic analysis.** The DNA of each CRKP isolate was purified and recovered by a silica gel column (D3146, HiPure Bacterial DNA kit) after incubation. Paired-end libraries with insert sizes of ∼300 bp were prepared following Illumina's standard genomic DNA library preparation procedure (VAHTS Universal DNA Library Prep kit for Illumina V3) and sequenced on an Illumina NovaSeq 6000 using the S4 reagent kits (v1.5) according to the manufacturer's instructions. The sequencing reads of each isolate were quality trimmed and assembled into contigs using EToKi (45). The ST, antibiotic resistance genes, and virulence determinants of each isolate were predicted based on its genomic assemblies using kleborate (46). All the isolates from ST11 were aligned using the EToKi align module onto a reference genome (GCF_011066505.1; sample from blood, Hong Kong, China, 2016) to obtain a multiple sequence alignment of the nonrepetitive, core genomic regions that were shared by ≥95% of genomes. The resulting alignments were subjected to a maximum-likelihood phylogeny by the EToKi phylo module, had the recombinant regions removed using RecHMM (47), and were visualized using GrapeTree online software (48). Similarly, we also aligned all the ST14 isolates onto a reference genome (GCF_001521895.1, sample from urine, Jiangxi, China, 2014) and estimated a maximum-likelihood phylogeny of ST14.

**Bayesian inference of the predominant populations.** We inferred the temporal and geographic origins of the ST11 and ST14 populations using BEAST v2.6.7 (49). For each ST, three Bayesian models were prepared with each of three population priors of "constant population," "extended skyline plot," and "birth-death skyline serial." In addition, all models used the GTR as the substitution model, the

optimized relaxed clock (50) as the substitution rate model, and the strict clock as the geographic transfer model. The Bayes factor of each model was summarized from 8 parallel nested sampling runs. For both STs, the model with constant population priors achieved the greatest Bayes factor, and was therefore used for population inferences. A set of posterior samples and trees with effective sampling sizes of 326 and 432 for ST11 and ST14 were generated from the nested sampling procedure above, and the maximum clade credibility trees were summarized using tree annotator and visualized using the ETE3 package in python.

**Identification of carbapenemase-encoding plasmids.** To detect plasmids in the SCMC isolates, we downloaded all 6,779 complete sequences of *Klebsiella* plasmids from the NCBI RefSeq database (July 2022). The assembly of each SCMC isolate was aligned onto the complete plasmids using BLASTn, and the alignment scores for each plasmid among the SCMC isolates were summarized. For each SCMC assembly, plasmids that were similar to the contigs encoding carbapenemase and obtained the greatest alignment scores were selected. The identified plasmids were aligned onto the NCBI RefSeq database again to obtain similar plasmids, and their phylogenies were estimated using the EToKi module. The incompatibility types and mobilities of the selected plasmids were determined using MOB-Typer software (51).

**Characterization of core genome and accessory genome.** To characterize the core and accessory genomes between "isolate pairs" from the same patient, we used the PEPPAN pipeline, which can reliably construct pangenomes from thousands of genetically diverse bacterial genomes (52). Based on the pangenome construction of 39 isolates, we developed a gene presence/absence matrix to calculate the genetic differences between "isolate pairs" using PEPPAN_parser. We identified the core genomes by PEPPAN_parser, which were present in over 95% of the genomes, and counted the number of allelic variations in core genomes. In addition, the genes that variably present among individual genomes were defined as accessory genes.

**Statistical analysis.** We used the chi-square test or Fisher exact test for categorical variables, and a Student *t* test or Mann-Whitney U-test was used for continuous variables. A *P* value of <0.05 was considered statistically significant, and all tests were two-sided. Statistical analyses were performed using the IBM SPSS Statistics (version 23.0) and WHONET (version 5.6).

**Data availability.** The raw sequence data reported here have been deposited in the Genome Sequence Archive (Genomics, Proteomics, and Bioinformatics 2021) in National Genomics Data Center (Nucleic Acids Research 2022), China National Center for Bioinformation/Beijing Institute of Genomics, Chinese Academic of Science (PRJCA012323, GSA CRA008499), and are publicly accessible (http://ngdc.cncb.ac.cn/gsa).

## SUPPLEMENTAL MATERIAL

Supplemental material is available online only.
**SUPPLEMENTAL FILE 1**, XLS file, 0.2 MB.
**SUPPLEMENTAL FILE 2**, XLS file, 0.1 MB.
**SUPPLEMENTAL FILE 3**, PDF file, 1.6 MB.

## ACKNOWLEDGMENTS

This research was supported by the 2021 Shanghai University teacher training plan/industry university research practice plan project (EYJ26.RL017), the National Natural Science Foundation of China (K116400121), and the Natural Science Foundation of Jiangsu Province (SL16400121).

We acknowledge Richard Stark and Sascha Ott for help with the maintenance and acquisition of data from the private *Klebsiella* database in EnteroBase. We also gratefully thank Fuping Hu and Yan Guo of the Institute of Antibiotics in Huashan Hospital (Fudan University) for technical assistance with the antimicrobial susceptibility testing.

Z.Z., Q.C., and X.W. contributed to the conception and design of the study. M.J., H.L., and X.L. performed the experimental work and N.S., Y.Z. and W.S. contributed to data acquisition. Z.Z., M.J., H.L., and X.L. participated in data analysis and manuscript writing. All authors read and approved the final manuscript. All authors contributed to the article and approved the submitted version.

We declare there are no conflicts of interest.

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
