## [Reviewer comments · Microbiology Spectrum]

Microbiology Spectrum

Genomic analysis revealed the international and domestic transmission of Carbapenem-Resistant *Klebsiella pneumoniae* in Chinese Pediatric Patients

Muxiu Jiang, Heng Li, Xiao Liu, Nan Shen, Yuanjie Zhou, Wenting Song, Xing Wang, Qing Cao, and Zhemin Zhou

Corresponding Author(s): Zhemin Zhou, Pasteurien College, Suzhou Medical College of Soochow University

Review Timeline:

Submission Date:	August 15, 2022
Editorial Decision:	October 5, 2022
Revision Received:	November 15, 2022
Editorial Decision:	November 26, 2022
Revision Received:	December 15, 2022
Accepted:	December 29, 2022

Editor: Rafael Vignoli

Reviewer(s): Disclosure of reviewer identity is with reference to reviewer comments included in decision letter(s). The following individuals involved in review of your submission have agreed to reveal their identity: Carolina Silva Nodari (Reviewer #2)

Transaction Report:

DOI: <https://doi.org/10.1128/spectrum.03213-22>

October 5, 2022

Prof. Zhemin Zhou
Pasteurien College, Suzhou Medical College of Soochow University
No. 199, Ren'ai Road, Suzhou
Suzhou, Jiangsu 215123
China

Re: Spectrum03213-22 (Genomic analysis revealed the international and domestic transmission of Carbapenem-Resistant *Klebsiella pneumoniae* in Chinese Pediatric Patients)

Dear Prof. Zhemin Zhou:

Thank you for submitting your manuscript to Microbiology Spectrum. Your article has been reviewed by two experts on the subject and both have agreed that although the article is interesting for publication, it requires some modifications. When submitting the revised version of your paper, please provide (1) point-by-point responses to the issues raised by the reviewers as file type "Response to Reviewers," not in your cover letter, and (2) a PDF file that indicates the changes from the original submission (by highlighting or underlining the changes) as file type "Marked Up Manuscript - For Review Only". Please use this link to submit your revised manuscript - we strongly recommend that you submit your paper within the next 60 days or reach out to me. Detailed instructions on submitting your revised paper are below.

Link Not Available

Sincerely,

Rafael Vignoli

Journals Department
Reviewer comments:

Reviewer #1 (Comments for the Author):

Very nice work and mainly well presented. My only general comment concerns the choice of only 57 isolates from blood stream infections for phenotypic AMR analyses. Otherwise, there is some need for checking language, some examples given below.
Lines 91-92; " were died" should be " died"
Lines 94-95; "were collected" written twice, only once needed
Line 167; " was originated" could be just "originated"
Line 171; "that kept being isolated" could be "that were continuously detected"

Line 185; "isolates resistant" should be "isolates were resistant"

Line 212; "co-exists" should be "co-existence"

Line 213; "than what expected" should be "than what was expected"

Lines 285-286; Sentence beginning "After excluding" is not complete and could be merged with the sentence following this one.

Lines 291ff; the long sentence on testing antimicrobial agents is not complete.

Reviewer #2 (Comments for the Author):

The manuscript by Jiang and colleagues describes the antimicrobial susceptibility profile and genetic relationship of *K. pneumoniae* causing infection in Chinese pediatric patients. This well-written study highlights transmission events of carbapenemase-producing isolates within China, as well as the presence of imported strains. They also present the most frequent plasmids associated with the dissemination of carbapenemase encoding genes. Below I present some comments and suggestions to improve the presentation of some results in order to make the data supporting some of the conclusions clearer.

- Lines 110-112: why was antimicrobial susceptibility testing only performed for isolates recovered from bloodstream samples?

- Lines 144-160: Authors mention in the first sentence of this section (lines 144-146) that only NDM-1-producing CRKP were included in the analysis. Yet, in line 153, they describe the introduction of the pSCMC3-IMP-4 plasmid. Shouldn't all ST14 CRKP be included in this analysis? If that is already the case, please rephrase the initial sentence to make it more clear. If not, consider expanding the analysis to all ST14 isolates evaluated. Finally, by observing Figure 3A, it is not possible to identify in which Chinese cities the different ST14 strains were introduced, as described in lines 151-153, since color legend only includes "China", and not the different regions/cities. Consider revising the Figure to help supporting these conclusions.

- Lines 169-171: It is not possible to clearly identify the years of isolation of the isolates belonging to the major cluster, as described in this sentence. Since there is no identification of the isolates in the tree, it is not possible to refer to the metadata presented in the Supplementary Dataset. Please revise.

- Lines 171-174: My understanding when evaluating the tree is that all isolates originated from either Zhejiang or Shanghai. Please provide additional information to support your conclusion.

- Lines 184-185: Consider including some discussion on the reason for the high CAZ-AVI MICs among KPC-2-producing isolates.

- Lines 211-214: Please include additional information and/or references describing how the "expected frequency of carbapenemase co-occurrence" was determined.

- Lines 229-232: Consider including some discussion on the possible sources of those plasmids.

- Lines 246-248: This cannot be confirmed by observing Figure 3B, since the isolate names are not presented in the tree. Please clarify or modify the Figure accordingly.

Minor comments:

- Lines 94-95: remove the "were collected" at the end of the sentence.

- Lines 101-103: include the percentage of NDM- and IMP-producing isolates, as it was done for KPC-2.

- Lines 105-109: Isolates from the same patient should be removed from the analysis, since they are clearly the same strain, as demonstrated in lines 107-109.

- Lines 123-124: Please remove *floR* from the sentence, since this gene encodes a MFS transporter conferring resistance to phenicols, and not quinolones.

- Lines 308-315: For both ST11 and ST14 reference genomes (GCF_011066505 and GCF_001521895), include some relevant metadata in the text, such as year and place of isolation, as well as biological source.

- Figure 2B: what do the percentages refer to? Please clarify.

- Figure 3: include a scale to indicate how many isolates are included in each branch or remove the pie charts to avoid confusion.

Staff Comments:

Preparing Revision Guidelines

- Point-by-point responses to the issues raised by the reviewers in a file named "Response to Reviewers," NOT IN YOUR COVER LETTER.
- Upload a compare copy of the manuscript (without figures) as a "Marked-Up Manuscript" file.
- Each figure must be uploaded as a separate file, and any multipanel figures must be assembled into one file.
- Manuscript: A .DOC version of the revised manuscript

- Figures: Editable, high-resolution, individual figure files are required at revision, TIFF or EPS files are preferred

Please return the manuscript within 60 days; if you cannot complete the modification within this time period, please contact me. If you do not wish to modify the manuscript and prefer to submit it to another journal, please notify me of your decision immediately so that the manuscript may be formally withdrawn from consideration by Microbiology Spectrum.

Dear reviewers,

-Thank you very much for your comments and professional advice concerning our manuscript entitled “Genomic analysis revealed the international and domestic transmission of Carbapenem-Resistant *Klebsiella pneumoniae* in Chinese Pediatric Patients” (Paper #Spectrum03213-22).

These comments are all valuable and very helpful for revising and improving the academic rigor of our article, as well as the important guiding significance to our research. We have studied comments carefully and have made correction which we hope meet with approval. Revised portion are marked in yellow in the paper. Furthermore, we would like to show the responds to comments as follow:

Reviewer 1#

Q1. My only general comment concerns the choice of only 57 isolates from blood stream infections for phenotypic AMR analyses.

-We highly appreciate the reviewer’s comments. We focused on the AMR phenotypes of blood-stream isolates due to their clinical significance and rapid disease progression. We suggested the use of genomic-based prediction for urgent management of blood-stream infections. CRKP isolates from other sources were less urgent and an accurate phenotypic test would be more appropriate.

In the study, the selection of isolates was expended several times, and 16 additional blood-stream isolates that did not have phenotypic data were included at the end. Here we added results of antimicrobial susceptibility testing for these additional blood stream isolates, and revised the main text (line108-119), as well as Figure 1D and Table 2 accordingly.

Q2. There is some need for checking language, some examples given below.

Lines 91-92; " were died" should be " died"

-Corrected.

Lines 94-95; "were collected" written twice, only once needed.

-Corrected.

Line 167; " was originated" could be just "originated".

-Corrected.

Line 171; "that kept being isolated" could be "that were continuously detected"

-Corrected.

Line 185; "isolates resistant" should be "isolates were resistant"

-Corrected.

Line 212; "co-exists" should be "co-existence"

-Corrected.

Line 213; "than what expected" should be "than what was expected"

-Corrected.

Lines 285-286; Sentence beginning "After excluding" is not complete and could be merged with the sentence following this one.

-Thanks very much. We have re-written the sentence.

It now reads: "A total of 198 CRKP isolates from 178 pediatric patients were included in the present study after excluding patients with poly-microbial infections or incomplete medical records". Thank again.

Lines 291; the long sentence on testing antimicrobial agents is not complete.

-Thanks very much for the comments. We have re-written the sentence.

It now reads: "The minimum inhibitory concentration (MIC) value for 17 antimicrobial agents (imipenem, meropenem, ceftazidime-avibactam, cefpiroxam, tigecycline, amikacin, polymyxin

B, ceftazidime, cefuroxime, aztreonam, ciprofloxacin, levofloxacin, cefoperazone-sulbactam, piperacillin-tazobactam, trimethoprim-sulfamethoxazole, cefazolin, ceftriaxone) were determined using broth microdilution method (Sensitire CHN1GNHH panel, Thermo Fisher Scientific) for 73 CRKP isolates” . Thanks again.

Reviewer 2#

Q1. Lines 110-112: why was antimicrobial susceptibility testing only performed for isolates recovered from bloodstream samples?

-We gratefully appreciate the reviewers' comments. The question has been answered in Q1 of Reviewer 1.

Q2. Lines 144-160: Authors mention in the first sentence of this section (lines 144-146) that only NDM-1-producing CRKP were included in the analysis. Yet, in line 153, they describe the introduction of the pSCMC3-IMP-4 plasmid. Shouldn't all ST14 CRKP be included in this analysis? If that is already the case, please rephrase the initial sentence to make it more clear. If not, consider expanding the analysis to all ST14 isolates evaluated. Finally, by observing Figure 3A, it is not possible to identify in which Chinese cities the different ST14 strains were introduced, as described in lines 151-153, since color legend only includes "China", and not the different regions/cities. Consider revising the Figure to help supporting these conclusions.

-We apologized for the mistake. We have revised the sentence to include all ST14 isolates. We have also revised figure 3B to highlight all ST14 isolates from Shanghai with stars, and add an arrow pointing to the ST14 isolate from Jiangxi, China. Figure legend has also been modified accordingly. Thank you.

Q3. Lines 169-171: It is not possible to clearly identify the years of isolation of the isolates belonging to the major cluster, as described in this sentence. Since there is no identification of the isolates in the tree, it is not possible to refer to the metadata presented in the Supplementary Dataset. Please revise.

-We apologize for this unclear description. We have now given out the raw tree file (Appendix 2) as well as associated metadata as supplementary Data 1. Thank you.

Q4. Lines 171-174: My understanding when evaluating the tree is that all isolates originated from either Zhejiang or Shanghai. Please provide additional information to support your conclusion.

-Thank you for pointing out this mis-use of words. We meant to describe the recent transmissions of SCMC ST11-CRKP isolates between provinces. We have revised this sentence to read “According to the tree, we could notice that isolates have been transferred between provinces frequently and many SCMC isolates were recently transferred from Zhejiang, Guangdong, Shandong, Sichuan, Beijing, and other Chinese provinces” on new line 166 to 169. Hope our response answered your question. Thanks again.

Q5. Lines 184-185: Consider including some discussion on the reason for the high CAZ-AVI MICs among KPC-2-producing isolates.

-Thanks for the kind suggestions. We have added related discussion on the new line 193 to 199. It now reads: “ In particular, 4 KPC-2 isolates exhibited ceftazidime-avibactam resistances, even though none of the patients have undergone any ceftazidime-avibactam treatment. Previous studies revealed that ceftazidime-avibactam resistance in KPC-producing isolates were associated with point mutations, increased gene copy number, and/or overexpression of the *bla*_{KPC-2} gene. However, we found that there was no additional point mutation, nor higher copy

numbers for the KPC-2 in the ceftazidime-avibactam resistant isolates comparing to their genetically related, susceptible neighbors. Therefore, the underlying mechanism for developing ceftazidime-avibactam resistance in pediatric patients is needed to illustrate”. Thanks again.

Q6. Lines 211-214: Please include additional information and/or references describing how the "expected frequency of carbapenemase co-occurrence" was determined.

-Thanks for this kind suggestion. We have added related discussion in the manuscript on the new line 237 to 241, and three references were added in the manuscript.

It now reads:“ Previous studies have reported one plasmid carrying two or more carbapenemase genes, which can probably extend their resistant profile. In recent years, the co-occurrence of *bla*_{KPC-2} and *bla*_{NDM-5} in a ST11 CRKP isolate, or the coexistence of *bla*_{IMP-4} and *bla*_{NDM-1} in one plasmid have been reported sporadically worldwide” .

References added in the manuscript

1. Hu R, Li Q, Zhang F, Ding M, Liu J, Zhou Y. 2021. Characterisation of *bla*_{NDM-5} and *bla*_{KPC-2} co-occurrence in KL64-ST11 carbapenem-resistant *Klebsiella pneumoniae*. J Glob Antimicrob Resist 27: 63-66.
2. Huang J, Zhang S, Zhao Z, Chen M, Cao Y, Li B. 2021. Acquisition of a Stable and Transferable *bla*_{NDM-5} -Positive Plasmid With Low Fitness Cost Leading to Ceftazidime/Avibactam Resistance in KPC-2-Producing *Klebsiella pneumoniae* During Treatment. Front Cell Infect Microbiol 11: 658070.
3. Xiao T, Peng K, Chen Q, Hou X, Huang W, Lv H, Yang X, Lei G, Li R. 2022. Coexistence of *tmexCD-toprJ*, *bla*_{NDM-5}, and *bla*_{IMP-4} in One Plasmid Carried by Clinical *Klebsiella spp*. Microbiol Spectr 10(3): e00549-22.

Q7. Lines 229-232: Consider including some discussion on the possible sources of those plasmids.

We appreciate it very much for this comment, and we have added two new paragraphs (lines 205- 228) on the sources of plasmids, and discussed their different patterns.

It now reads: “We reconstructed sequences of four dominant plasmids in this study, and compared each plasmid with genetically similar plasmids from NCBI RefSeq database to estimate a maximum-likelihood phylogeny. The results shown that the plasmids fell into two categories of “dispersed” and “restricted” in terms of both host and geographic distributions. pSCMC1 (IncX3) and pSCMC2 (IncN) both carried mostly *bla*_{NDM} genes and had “dispersed” host and country sources (supplementary Figure 3, Figure 2C-F). pSCMC1 was carried by bacteria from >20 *Enterobacteriaceae* species in 35 countries globally (Figure 2C), and pSCMC3 (IncN) plasmids were found in 23 *Enterobacteriaceae* species from 16 countries (Figure 2E, supplementary Figure 3). Both plasmids transmitted between different hosts and different countries very frequently, leading to the reports of genetically almost identical plasmids from multiple species in many countries (pie-charts in Figures 2C and E and supplementary Figure 3). As a result, it is impractical to estimate the origin of the plasmid using sequences alone. In contrast, pSCMC2 (IncFII) and pSCMC4 (IncU) were “restricted” and mostly only found in *Klebsiella* from China (Figure 2D and F, supplementary Figure 3). Most pSCMC4 plasmids carried *bla*_{NDM}, whereas pSCMC2 plasmids carried mostly *bla*_{KPC-2} genes and were associated with the national dissemination of ST11-KPC-2 CRKPs (25).

It is very concerning that almost identical plasmids could be isolated from epidemiological unrelated sources. Previous studies also reported the finding of genetically closely related plasmids between those from humans and from environments, such as hospitals, farm yard and

aquatic environment, etc. (26-29). Therefore, using of measurements other than anti-microbials, e.g., hand hygiene, patient isolations, minimization of physical contact with animals and disinfection of environment, may decrease the spread of *bla*_{NDM} and *bla*_{IMP} genes among children. ”

Q8. Lines 246-248: This cannot be confirmed by observing Figure 3B, since the isolate names are not presented in the tree. Please clarify or modify the Figure accordingly.

-Thanks for your kind suggestions. We pointed out the two isolates (HB021261 and HB021722) with arrows in Figure 3B. And we highlighted them in supplementary Data 1 to make this description more clear. Thanks again.

Minor comments:

Q9. -Lines 94-95: remove the "were collected" at the end of the sentence.

-Corrected.

Q10. Lines 101-103: include the percentage of NDM- and IMP-producing isolates, as it was done for KPC-2.

- Thanks for your kind suggestions. We have revised this sentence to “NDM variants (96, 48.5%) were the primary carbapenemases which carried by 96 isolates from 37 STs, followed by KPC-2 (53, 26.7%) that were mostly from ST11, and IMP variants (33, 16.7%) from 19 STs”, here we adjusted the proportions after subtracting 8 strains containing both carbapenemases. Please check the new line 94 to 96. Thanks again.

Q11. Lines 105-109: Isolates from the same patient should be removed from the analysis, since they are clearly the same strain, as demonstrated in lines 107-109.

- Thanks for the suggestions. However, we would like to keep these CRKP isolates from the same patient. Mixed-infections and *in vivo* evolution of pathogens have been widely reported,

and found to be clinically important for choices of AMRs for treatments, especially in prolonged infections. We believe that it is very important to systematically evaluate this problem, which have not been done properly in CRKP.

To discuss the problem in greater detail, we now included a new paragraph (lines 286-306). We found that, while the core genome of isolates from the same patient was highly conserved, the accessory genome had changed relatively fast.

It now reads: “Furthermore, genetically different strains from the same species could be isolated from a same patient. Such micro-diversity was a result of either repetitive infections, or *in vivo* diversifying evolutions of a single clone during prolonged infections, and could be clinically important when different strains had different AMR susceptibilities. For example, ST11-KPC-2 isolates have been reported to gain additional resistances to ceftazidime-avibactam and tigecycline by mutations in D179Y and the *ramR* gene in long-term treatments (20, 39). Here we compared genomic sequences of 19 sets of CRKPs that were each from the same patients. We found repetitive infections of different strains that differed each other by >3400 core gene sequences variations and >1500 gain/loss of accessory genes for two patients (supplementary Data 2). Detailed investigation shown that one patient carried different CRKPs in different hospital admissions, and the other gained a second infection during her >1-year stay in SCMC. In contrast, we found only 1.3 (0-4) mutations in the core genes for strains from the remaining 17 patients (supplementary Data 2), indicating *in vivo* evolution of single infections. Interestingly, much larger gene content variations (~31 genes; 0-283) were found in their accessory genomes, which were associated with gain/loss of prophages and plasmids. Similar phenomenon has also been reported in *Salmonella*, where a lab-leaking strain gain multiple prophages and plasmids without any trackable change in the core genome (40). It also

highlighted the roles of AMRs and virulence factors carried by plasmids, prophages and other mobile elements as the main driving forces for *in vivo* adaptation of CRKP.”

Q12. Lines 123-124: Please remove floR from the sentence, since this gene encodes a MFS transporter conferring resistance to phenicols, and not quinolones.

-We apologized for this mistake. We have revised figure 1D accordingly. Thank you.

Q13. Lines 308-315: For both ST11 and ST14 reference genomes (GCF_011066505 and GCF_001521895), include some relevant metadata in the text, such as year and place of isolation, as well as biological source.

-Thanks for the kind suggestions. Related information has been added in this manuscript. The ST11 reference genome, GCF_011066505, which isolated from bloodstream from an inpatient living in Hong Kong, China, in 2016. And the ST14 reference genome, GCF_001521895, which recovered from a urine sample from a 65-year-old female patient hospitalized in the teaching hospital of Nanchang University, from Jiangxi, China, in 2015. Thanks again.

Q14. Figure 2B: what do the percentages refer to? Please clarify.

-We apologized for this unclear description. The percentages of Figure 2B refer to the number of isolates from bloodstream to all sources isolates for each pSCMC plasmid. For example, a total of 64 pSCMC1-carrying CRKP isolates, of which 37 isolates were originated from bloodstream infections, which approximately accounting for 58% (37/64) as described in figure 2B. We have revise the sentence in the figure legend part to make it clear.

In now reads: “ The percentages refer to the number of isolates (top) and STs (bottom) from bloodstream to all sources isolates for each pSCMC plasmid.”

Q15. Figure 3: include a scale to indicate how many isolates are included in each branch or remove the pie charts to avoid confusion.

-Thanks for the suggestions. We have added detail information in supplementary Data 1 to describe these isolates included in each branch in Figure 3, and the raw data related MCC trees for ST14 and ST11 isolates in Figure 3 can be available in Appendix 1 and Appendix 2. Thanks again!

November 25, 2022

Prof. Zhemin Zhou
Pasteurien College, Suzhou Medical College of Soochow University
Pasteurien College
No. 199, Ren'ai Road, Suzhou
Suzhou, Jiangsu 215123
China

Re: Spectrum03213-22R1 (Genomic analysis revealed the international and domestic transmission of Carbapenem-Resistant *Klebsiella pneumoniae* in Chinese Pediatric Patients)

Dear Prof. Zhemin Zhou:

Thank you for submitting your manuscript to Microbiology Spectrum. Although the reviewers understand that the paper has been improved in the new version, there are still some important points to be resolved. When submitting the revised version of your paper, please provide (1) point-by-point responses to the issues raised by the reviewers as file type "Response to Reviewers," not in your cover letter, and (2) a PDF file that indicates the changes from the original submission (by highlighting or underlining the changes) as file type "Marked Up Manuscript - For Review Only". Please use this link to submit your revised manuscript - we strongly recommend that you submit your paper within the next 60 days or reach out to me. Detailed instructions on submitting your revised paper are below.

Link Not Available

Sincerely,

Rafael Vignoli

Journals Department
Reviewer comments:

Reviewer #1 (Comments for the Author):

To get the language better you should consult an expert in English, some of the texts require some attention.

Reviewer #2 (Comments for the Author):

I appreciate the efforts made to improve the manuscript. However, some information are still missing or are not made clear

enough. Unfortunately, the added Appendixes and Metadata table did not improve visualization, on the contrary, they require that readers manually label more than 400 isolates, not to mention that the distinct tree topologies presented in Figure 3 and the Appendixes make it impossible to quickly identify branches of interest. Below I present additional comments to some of the Authors' answers, which follow the same numbering as presented in the rebuttal letter.

- Q3: Information is still not clear, as it requires readers to manually match the years of isolation presented in the metadata table with the tree branches. I was also not able to evaluate "supplementary Data 1" (the file seems to have been corrupted at some point, making it impossible to open from my side), making it even harder to evaluate the new information added. Ideally, this could be improved by making the dataset (and its metadata) available as a workspace at Enterobase, but my understanding is that their *Klebsiella* database is still not publicly available. Therefore, authors should consider adding colors labels referring to isolation years on the branch tips of the tree presented in Appendix 1. This can be easily done with iTOL, for example.

- Q4: Unfortunately the rewording did not make the information clear, since my question referred to the fact that I am not able to easily identify those transmission events. I would appreciate if authors would highlight them somehow in the tree, maybe using arrows to point out to which branches they referred to when suggesting that were originated from other provinces.

- Q6: In my opinion, the sporadic reports mentioned do not justify the expectation that carbapenemase co-occurrence should be more frequent, since it rarely offers an evolutionary advantage to the bacterial strain, and should remain rare. I strongly advise removing this sentence, as it can make readers believe that such conclusion was supported by some statistical analysis, which does not seem to have been the case.

- Q11: Authors mention that there was important variation in the accessory genome between "isolate pairs" from the same patient. However, there is no description of how the accessory genome was evaluated, only the results are made available in Supplemental Dataset 2. Please add the methods used for the evaluation of accessory genome (not no mention how gene annotation was performed to identify the CDSs belonging to core genome, since the data presented refers to allelic variation, and not SNPs) in the revised manuscript. Was a cgMLST scheme used for this evaluation? Please describe.

- Q15: As mentioned in my comment to the response to Q3, the new appendixes and metadata table do not improve data visualization. Please revise the Figures as previously suggested.

Staff Comments:

Preparing Revision Guidelines

Please return the manuscript within 60 days; if you cannot complete the modification within this time period, please contact me. If you do not wish to modify the manuscript and prefer to submit it to another journal, please notify me of your decision immediately so that the manuscript may be formally withdrawn from consideration by Microbiology Spectrum.

Reviewer 1#

To get the language better you should consult an expert in English, some of the texts require some attention.

-Thank you for the comments. We revised the manuscript considerably and sent the manuscript to AJE for English editing. The corrected details are marked in yellow in the paper. We hope the revised manuscript could be acceptable for you.

Reviewer 2#

The added Appendixes and Metadata table did not improve visualization, on the contrary, they require that readers manually label more than 400 isolates, not to mention that the distinct tree topologies presented in Figure 3 and the Appendixes make it impossible to quickly identify branches of interest.

-Thanks for the comments. We redrew the Figure 3 in iTOL v3.0 and revised the figure legends. The updated MCC trees of ST14 and ST11 were presented in Figure 3 and Figure 4, respectively. We hope that the revised figure has addressed your suggestions and meet with approval.

Below I present additional comments to some of the Authors' answers, which follow the same numbering as presented in the rebuttal letter.

- Q3: Information is still not clear, as it requires readers to manually match the years of isolation presented in the metadata table with the tree branches. I was also not able to evaluate "supplementary Data 1" (the file seems to have been corrupted at some point, making it impossible to open from my side), making it

even harder to evaluate the new information added. Ideally, this could be improved by making the dataset (and its metadata) available as a workspace at Enterobase, but my understanding is that their Klebsiella database is still not publicly available. Therefore, authors should consider adding colors labels referring to isolation years on the branch tips of the tree presented in Appendix 1. This can be easily done with iTOL, for example.

-Thanks for the suggestions. We have revised the figure in iTOL. The new Figure 4 showed the years of isolation and the geographical origin of ST11 isolates clearly. Additionally, we apologized for the corrupted file, and we have re-uploaded the supplementary Data 1 to address the issues.

- Q4: Unfortunately the rewording did not make the information clear, since my question referred to the fact that I am not able to easily identify those transmission events. I would appreciate if authors would highlight them somehow in the tree, maybe using arrows to point out to which branches they referred to when suggesting that were originated from other provinces.

-We apologized for this unclear description. We pointed out these isolates referring to transmission events with arrows in Figure 4. We also revised the following sentence as “According to the tree, isolates have been frequently transferred between provinces, and many SCMC isolates were recently transferred from Fujian, Guangdong, Shandong, Henan, and other areas of China”. Hope our response answered your question. Thanks you.

- Q6: In my opinion, the sporadic reports mentioned do not justify the expectation that carbapenemase co-occurrence should be more frequent, since it rarely offers an evolutionary advantage to the bacterial strain, and should remain rare. I strongly advise removing this sentence, as it can make readers believe that such conclusion was supported by some statistical analysis, which does not seem to have been the case.

-Thanks for your kind suggestions. After several rounds of discussion by all authors, we decided to remove this sentence as suggested by reviewer. The present paragraph were showed as: "In addition, CRKP isolates that carry multiple carbapenemases in one or multiple plasmids have been widely reported and studied (7, 30, 31). We also found 8 isolates that encoded two carbapenemases, IMP-4 and NDM-1 or KPC-2 and NDM-5. Several studies have revealed that the convergence of *bla*_{KPC-2} and *bla*_{NDM-5} or the coexistence of *bla*_{IMP-4} and *bla*_{NDM-1} in a single strain has the ability to extend the resistance profile of the bacteria, allowing the bacteria to potentially be transmitted between patients (32-34), posing a significant threat to public health. Therefore, it is necessary to strengthen the surveillance of these isolates among children to improve clinical treatment and management." Please check the new line 243-251. Thanks again.

- Q11: Authors mention that there was important variation in the accessory genome between "isolate pairs" from the same patient. However, there is no description of how the accessory genome was evaluated, only the results are made available in Supplemental Dataset 2. Please add the methods used for the evaluation of accessory genome (not no mention how gene annotation was performed to identify the CDSs belonging to core genome, since the data

presented refers to allelic variation, and not SNPs in the revised manuscript.

Was a cgMLST scheme used for this evaluation? Please describe.

-We apologized for this missing description. We added related description in the method part as follows “To characterize the core and accessory genomes between "isolate pairs" from the same patient, we used the PEPPAN pipeline, which can reliably construct pangenomes from thousands of genetically diverse bacterial genomes (53). Based on the pangenome construction of 39 isolates, we developed a gene presence/absence matrix to calculate the genetic differences between "isolate pairs" using PEPPAN_parser. We identified the core genomes by PEPPAN_parser, which were present in over 95% of the genomes, and counted the number of allelic variations in core genomes. In addition, the genes that variably present among individual genomes were defined as accessory genes. ” Thank you.

References added in the manuscript

1. Zhou Z, Charlesworth J, Achtman M. Accurate reconstruction of bacterial pan- and core genomes with PEPPAN. *Genome Res.* 2020. 30(11):1667-1679.

- Q15: As mentioned in my comment to the response to Q3, the new appendixes and metadata table do not improve data visualization. Please revise the Figures as previously suggested.

-We gratefully appreciate the reviewers’ comments. The Figure 3 in our original submission has been revised as mentioned above. Please check the new Figure 3 and

Figure 4. We hope that the modification has solved the problem you mentioned.

Thanks again!

December 29, 2022

Prof. Zhemin Zhou
Pasteurien College, Suzhou Medical College of Soochow University
Pasteurien College
No. 199, Ren'ai Road, Suzhou
Suzhou, Jiangsu 215123
China

Re: Spectrum03213-22R2 (Genomic analysis revealed the international and domestic transmission of Carbapenem-Resistant *Klebsiella pneumoniae* in Chinese Pediatric Patients)

Dear Prof. Zhemin Zhou:

It is a pleasure for me to inform you that your manuscript has been accepted, and I am forwarding it to the ASM Journals Department for publication. You will be notified when your proofs are ready to be viewed.

Sincerely,

Rafael Vignoli
Editor, Microbiology Spectrum
